# Effectiveness of an Immersive Virtual Reality Simulation Speak-Up Training Program for Patient Safety in Novice Nurses: A Quasi-Experimental Study

**DOI:** 10.3390/healthcare13192435

**Published:** 2025-09-25

**Authors:** Hea-Kung Hur, Ji-Hea Choi, Ji-Soo Jung

**Affiliations:** Department of Nursing, Yonsei University Wonju College of Nursing, Wonju 26426, Republic of Korea; hhk0384@yonsei.ac.kr (H.-K.H.); papaya1978@naver.com (J.-S.J.)

**Keywords:** assertiveness, communication, nurses, patient safety, quasi-experimental study, simulation training, virtual reality

## Abstract

**Background/Objectives**: Novice nurses often struggle to assertively voice patient safety concerns due to limited clinical experience and hierarchical healthcare environments. Immersive virtual reality simulation (IVRS) may provide opportunities to practice assertive communication skills essential for interprofessional collaboration in a psychologically safe environment. This study aimed to evaluate the short- and long-term effectiveness of an IVRS speak-up training program in enhancing communication clarity and collaborative attitudes, as well as reducing experiences of nursing malpractice among novice nurses. **Methods**: A quasi-experimental study was conducted with 36 novice nurses (18 participants each in control and experimental groups). The experimental group received a 200 min IVRS speak-up training program incorporating TeamSTEPPS communication strategies through four virtual reality scenarios. The control group received a 110 min conventional group lectures and discussions on communication training covering the same TeamSTEPPS strategies. Communication clarity and collaborative attitudes were measured at baseline, immediately post-intervention, and 6 weeks post-intervention. Nursing malpractice experiences were assessed at 6 weeks post-intervention. **Results**: Communication clarity showed no significant interaction effect between groups and time points (F = 0.84, *p* = 0.437), though both groups demonstrated immediate post-intervention improvements. Collaborative attitudes showed a significant interaction effect (F = 4.23, *p* = 0.020), with the experimental group exhibiting significantly greater and sustained improvements at immediate and 6-week follow-up compared with the control group. No significant difference in nursing malpractice experiences was observed between groups (Z = 0.16, *p* = 0.872). **Conclusions**: The IVRS speak-up training program effectively enhanced novice nurses’ assertive communication skills in immersive, interactive environments tailored for realistic practice compared to conventional group discussion-based training. This program improved communication clarity in the short term and enhanced collaborative attitudes up to 6 weeks. Integrating or boosting IVRS training into novice nurses’ communication education may foster interprofessional collaboration and advance patient safety in clinical practice.

## 1. Introduction

Speak-up communication is a key factor in enhancing patient safety and preventing medical errors within clinical settings as well as in fostering effective interprofessional collaboration [1]. Often conceptualized as a form of assertive communication, speak-up behavior involves expressing concerns, sharing clinical observations, and advocating for patient safety without hesitation [1,2]. It empowers healthcare professionals to raise potential safety risks and promotes a culture of openness that reduces the likelihood of adverse events [3].

However, recent evidence indicates persistent barriers to effective speak-up communication and interprofessional collaboration in clinical practice. Studies consistently show that healthcare professionals, particularly nurses, often hesitate to speak up in hierarchical clinical environments, which undermines collaboration between physicians and nurses [4,5]. A systematic review by Lee et al. [5] revealed that hierarchical organizational culture remains a significant barrier preventing East Asian nurses from speaking up regarding patient safety, leading them to prefer formal reporting systems over direct communication. Such reliance on indirect communication mechanisms can delay timely interprofessional communication and compromise collaborative decision-making essential for optimal patient outcomes. Similarly, Jeong & Kim [6] identified that South Korean nurses primarily relied on patient safety reporting systems rather than direct speak-up communication, highlighting how communication barriers can inhibit interprofessional teamwork and delay critical safety interventions.

Moreover, novice nurses face unique challenges in speak-up communication because of poor confidence, fear of reprisal, and insufficient training, which limit their ability to engage effectively in interprofessional collaboration [4]. They are particularly vulnerable to communication barriers within hierarchical hospital organizational cultures, leading to underreporting of potential safety issues and breakdowns in collaborative care processes. Recent qualitative research suggests that one-time training programs for newly graduated nurses are insufficient for building sustained communication competencies, which support effective interprofessional teamwork; thus, ongoing mentoring is needed to develop speak-up capabilities and strengthen collaborative attitudes [6,7].

Existing educational programs often lack the support and training structures needed for novice nurses to effectively practice speak-up communication and build interprofessional collaboration skills in real-world clinical settings [1]. Traditional educational approaches typically provide one-time training sessions, which cannot adequately address the ongoing need for skill reinforcement and confidence building for effective teamwork. This gap is concerning, as evidence indicates that nurses’ communication skills can decline within a few months without continued practice [8].

Virtual reality simulation (VRS) has emerged as a promising alternative to traditional educational methods, providing learners with immersive and realistic environments for skill acquisition [9]. Immersive VRS (IVRS), which uses head-mounted displays (HMDs) and controllers, enables interactive training in 3D virtual spaces. These platforms offer safe, controlled spaces for repetitive, self-paced learning that can simulate complex interprofessional scenarios. A recent study demonstrated that IVRS-based TeamSTEPPS training markedly enhanced healthcare teams’ safety behaviors, with 90% of their measured safety behaviors, including comfort with speaking up and asking questions within interprofessional team contexts, significantly improved [10].

Integrating VRS with structured communication frameworks, such as TeamSTEPPS program, may further enhance nurses’ ability to communicate effectively in interprofessional settings and promote collaborative attitudes with physicians. The TeamSTEPPS program incorporates evidence-based communication techniques including ISBAR (Identification, Situation, Background, Assessment, Recommendation for structured reporting communication), check-back (closed-loop verification of received information), two-challenge rule (assertive challenge when initial concerns are ignored), and CUS (Concerned, Uncomfortable, Safety issue for escalating patient or staff safety concerns) [11,12,13]. However, while VRS has proven effective for communication training in healthcare teams, reports on the effectiveness of IVRS-based speak-up training for novice nurses remain limited. Furthermore, one-time training sessions are insufficient for developing speak-up capabilities among novice nurses [6,14], and repeated education is recommended to sustain long-term effects [8]. Therefore, establishing evidence on the sustained impact of IVRS in enhancing speak-up communication skills, particularly in interprofessional collaboration, is essential.

This study aimed to evaluate the short- and long-term effectiveness of an IVRS-based speak-up training program incorporating multiple patient care scenarios. Adapted from a previously validated IVRS program that enhanced team communication and collaboration among nursing students [15], this novel program was tailored for novice nurses in clinical practice. While the original program demonstrated significant short-term improvements in communication and collaborative attitudes among undergraduate nursing students, the present study examined whether these effects could be sustained among practicing novice nurses. Communication competencies, interprofessional collaboration skills, and communication-related nursing incidents were evaluated through repeated outcome assessments over a 6-week period.

We hypothesized that novice nurses who received the adapted IVRS speak-up training program would demonstrate significantly higher mean scores for communication clarity and collaborative attitudes compared with the control group receiving conventional group lectures and discussions on communication across all measurement time points and would report lower levels of communication-related nursing malpractice within interprofessional team contexts during the follow-up period.

## 2. Materials and Methods

### 2.1. Study Design

This study used a quasi-experimental design with non-equivalent control groups, using pre- and post-tests to assess the impact of the IVRS speak-up training program on patient safety. The study focused on its effects on communication clarity, collaborative attitudes, and nursing malpractice experience among novice nurses.

### 2.2. IVRS Speak-Up Program

#### 2.2.1. Development Process of the IVRS Speak-Up Training Program

The IVRS speak-up training program for novice nurses was adapted from a previously validated IVRS program originally designed to enhance teamwork communication and collaboration among undergraduate nursing students [15]. The original program aimed to develop standardized effective communication strategies (ISBAR and check-back) and mutually supportive communication techniques (two-challenge rule and CUS) through two inpatient scenario-based training modules. Scenario 1 featured a 54-year-old male patient undergoing partial gastrectomy, while scenario 2 involved a 55-year-old male patient with liver cirrhosis and ascites. Each scenario comprised two progressive learning stages: one focused on mild clinical situations for practicing ISBAR and check-back, and the other focused on more complex situations for applying the two-challenge rule and CUS [15]. The program was developed using the Oculus Quest 2 HMD via the Unity 3D platform (Meta Platforms, Inc., Menlo Park, CA, USA, 2020) and was aligned with the International Nursing Association for Clinical Simulation and Learning (INACSL) guidelines [16]. Communication techniques were structured according to the TeamSTEPPS^®^ 3.0 framework (Agency for Healthcare Research and Quality, Rockville, MD, USA) [11].

In this study, given the differences in clinical experience and role expectations between student and novice nurses, the original program was modified to reflect the realities of clinical nursing practice. The revised IVRS program for novice nurses particularly aimed to enhance assertive communication behaviors by promoting the application of mutually supportive communication strategies in actual interprofessional interactions. The primary goal of the revision was to strengthen novice nurses’ ability to express patient safety concerns assertively while maintaining professional respect for physicians. While retaining learning objective 1 (mastery of ISBAR and check-back), we also emphasized learning objective 2 (proficient use of the two-challenge rule) and learning objective 3 (proficient use of the CUS technique). To reinforce these objectives, learners were guided to practice both with and without the two-challenge rule and CUS techniques after reporting a patient’s condition. Following development, expert validation was conducted with three clinical nursing leaders (an education team leader, a ward team leader, and a section leader) and a professional VR developer. Feedback focused on alignment with the modified objectives, scenario realism, and user interaction design. To improve immersion and intuitiveness, visual and auditory feedback cues were enhanced, time constraints on communication steps were removed, and patient information screen text was enlarged. Quiz-like cues were also added to support skill acquisition. The program’s usability was then evaluated by three novice nurses (employed for <3 months at a tertiary hospital), using the 10-item System Usability Scale (SUS) [17]. The program received a mean score of 93.3 out of 100, indicating high usability. Participant feedback noted discomfort due to screen glare during transitions, which was addressed by implementing a fade-to-black effect to improve visual comfort. Through this iterative development process, the IVRS speak-up training program was finalized.

#### 2.2.2. Description of the IVRS Speak-Up Training Program

The IVRS speak-up training program was structured into four phases (preparation, briefing, IVRS practice, and debriefing of four multi-patient scenarios) in alignment with the INACSL Standards of Best Practice [16], with a total duration of 200 min.

Figure 1 provides a schematic overview of the program structure.

The preparation phase (40 min) introduced the significance of interprofessional collaboration and effective communication for patient safety. This phase included detailed instruction on standardized and supportive communication strategies, particularly ISBAR, check-back, two-challenge rule, and CUS technique. The subsequent briefing phase (40 min) was conducted immediately before the simulation, focusing on reviewing the training objectives, explaining scenario structure, and providing hands-on orientation to VR equipment and operational procedures. Through this phase, participants would understand how to respond appropriately within each scenario and were adequately prepared for the immersive learning experience.

The IVRS practice and debriefing of four multi-patient scenarios phases were delivered as integrated sessions, with debriefing conducted immediately following each simulation experience. To achieve learning objective 1, mild case scenarios 1-1 (patient with partial gastrectomy complaining of mild abdominal pain) and 2-1 (patients with liver cirrhosis and ascites with mild dyspnea) were used to provide foundational training on standardized effective communication techniques (ISBAR and check-back). Each scenario included an IVRS practice session (5 min) followed by a debriefing session (10 min). To achieve learning objective 2, severe case scenario 1-2 (patient with partial gastrectomy complaining of severe dyspnea) was used to train participants in the two-challenge rule as a mutually supportive communication strategy. This session included an IVRS practice session (10 min) followed by a debriefing session (20 min). Finally, to achieve learning objective 3, severe case scenario 2-2 (patients with liver cirrhosis and ascites with loss of consciousness) was used to train participants in both the two-challenge rule and CUS as mutually supportive communication strategies. This session included an IVRS practice session (15 min) followed by a debriefing session (30 min).

Figure 2 depicts the sample screens for the virtual environment. And, Figure 3 shows the operational workflow of the two-challenge rule in learning objective 2 of scenario 1-2. 

### 2.3. Participants and Setting

Convenience sampling was used to recruit novice nurses from a tertiary hospital (Wonju Severance Christian Hospital) of Yonsei University in Wonju City, South Korea. Sample size estimation was conducted using G*Power 3.1.9.4 (Heinrich Heine University, Düsseldorf, Germany), with a significance level of 0.05, statistical power of 0.80, and an effect size of 0.62 based on prior research [18]. Results indicated that a minimum of 32 participants would be required to detect differences in repeated measures between the experimental and control groups. To account for a potential 10% dropout rate, the final target sample size was set at 36, with 18 participants allocated to each group.

Eligibility criteria included novice nurses who had completed their job training within the past 3 months; demonstrated an understanding of the study’s purpose, methods, and procedures; and provided voluntary informed consent to participate. The exclusion criteria were those expected to experience motion sickness, a side effect of using an HMD required for IVRS applications [19]. Nurses with related symptoms, such as eye discomfort, nausea or dizziness, anxiety or claustrophobia, cardiovascular disease, and mental illness as well as those under certain medications (e.g., anti-anxiety medications, motion sickness medications), were also excluded. Moreover, to accurately identify the characteristics of novice nurses, those who had previously worked at primary or secondary hospitals were excluded. Predefined dropout criteria included withdrawal of consent, occurrence of severe VR-related symptoms (e.g., persistent nausea, dizziness, or eye strain lasting >30 min), inability to complete the full intervention protocol, and failure to attend scheduled sessions. Within the study period, no participants met either of these criteria.

Participants were recruited through an open recruitment notice targeting nurses who had been newly employed for 3 months. Novice nurses who wished to participate in the study contacted the researcher through the number listed in the recruitment notice. They then participated in the study at the simulation laboratory located within Yonsei University Wonju College of Nursing. Study participants were sequentially assigned to the control group until 18 individuals were included, after which the remaining participants were allocated to the experimental group until the total number of participants reached 36. Finally, all 36 participants completed the study without any dropouts in either the control or experimental groups, as no participants met any of the predefined dropout criteria during the study period.

### 2.4. Measurements

Participant characteristics, such as sex, age, clinical experience, and prior exposure to virtual reality learning, were collected to confirm baseline comparability between the experimental and control groups. The effectiveness of the IVRS speak-up training program was assessed in terms of communication clarity, collaborative attitudes, and experiences of nursing malpractice by participants at designated time points.

#### 2.4.1. Communication Clarity

Communication clarity, defined as “ensuring that the conveyed message is easily understood by others,” was measured using the Communication Clarity Scale originally developed by Marshall et al. [13] and translated into Korean by Cho [20]. The original scale was designed as an observer-rated assessment tool [13]; however, the Korean version used in this study was adapted by Cho [20] as a self-assessment instrument where participants evaluate their own communication performance. The scale consists of 14 items and is scored on a 5-point Likert scale, ranging from “not at all” (1 point) to “very much” (5 points), with higher scores indicating higher communication clarity. Item means were used for analysis. Cronbach’s alpha value for reliability was 0.77 in the study by Cho [20] and 0.93 in this study.

#### 2.4.2. Collaborative Attitudes

Collaborative attitudes, defined as “perceptions and beliefs about the importance of teamwork, shared decision-making, and mutual respect between physicians and nurses in delivering patient care,” was measured through self-assessment by participants using the Jefferson Scale of Attitudes Toward Physician–Nurse Collaboration, revised and supplemented by Hojat et al. [21] and translated to Korean by Lee & Sohn [12]. The scale consists of 15 items, namely sharing education and collaboration (7 items), care versus treatment (3 items), nurses’ autonomy (3 items), and physicians’ authority (2 items), and was scored on a 4-point Likert scale ranging from “not at all” (1 point) to “very much” (4 points), with higher scores indicating a more positive collaborative attitudes toward physician–nurse collaboration. Item means were used in this study. Cronbach’s alpha values were 0.80 in the study of Lee & Sohn [12] and 0.79 in this study.

#### 2.4.3. Experience of Nursing Malpractice

Experience of nursing malpractice was defined as “experience of making a mistake or recognizing nursing malpractice that could unexpectedly harm a patient while performing nursing duties.” The number of incidents experienced, including near-misses, was measured through self-assessment by participants using a structured questionnaire. This variable was included based on the theoretical premise that improved speak-up communication skills would enable novice nurses to voice safety concerns more effectively, potentially reducing communication-related errors and malpractice incidents through enhanced interprofessional collaboration and timely intervention. The degree of nursing malpractice experience was calculated as the incidence of experience × the number of times experienced, with no experience receiving 0 points and experience receiving 1 point. Higher scores indicate a higher degree of nursing malpractice experience. Participants were assessed at 6 weeks post-intervention to allow sufficient time for the communication training effects to manifest in clinical practice and influence safety-related outcomes.

### 2.5. Comparison with the Control Group

To minimize the risk of contamination between the control and experimental groups, intervention and data collection were completed first for the control group then for the experimental group. For comparison, the control group received a standardized traditional educational intervention designed to provide equivalent content coverage while using conventional teaching methods. The control group intervention consisted of three structured components delivered over 110 min by the same research assistant using standardized protocols. First, participants received a 40 min group lecture (4–6 participants per group) on effective communication techniques, including ISBAR, check-back, the two-challenge rule, and CUS, using identical theoretical content as the experimental group but delivered through traditional presentation methods. Second, participants viewed a 10 min video clip illustrating a physician–nurse communication scenario that demonstrated practical application of the taught communication techniques. Third, participants engaged in a 60 min facilitated group discussion where they analyzed the communication skills demonstrated in the video scenario using structured questions to ensure consistency across groups. This methodology provided equivalent theoretical knowledge and reflection opportunities while lacking the immersive, repetitive practice and individualized feedback characteristics of the IVRS intervention, ensuring a fair comparison between traditional and innovative educational approaches. All sessions were conducted in the same simulation laboratory setting to control for environmental factors. For ethical considerations, the participants in the control group were asked about their willingness to participate in the IVRS speak-up training program for patient safety after completing the control group training. Those who were willing were individually contacted and provided with the program after collecting all the data for the experimental group.

### 2.6. Implementation of the IVRS Speak-Up Training Program in the Experimental Group

The IVRS training program was administered by a trained research assistant who had completed a standardized training protocol covering VR equipment operation, scenario management, debriefing using the Gather–Analyze–Summarize (GAS) model [16], and emergency response procedures for VR-related adverse effects. To implement the IVRS speak-up training program in the experimental group, a chair with back support was set up in an empty classroom at the simulation center of Yonsei University Wonju College of Nursing. The space around the seat was cleared by removing obstructions around it to prevent falls or collisions. Prior to the intervention, a trained research assistant assessed each participant’s physical condition using a standardized checklist that included fatigue, muscle pain, skin irritation, eye strain, and dizziness. The assistant also monitored participants for any signs of discomfort after each simulation session. The intervention was delivered to all participants in the experimental group by the same assistant in accordance with the standardized procedure presented in Figure 1.

Through standardized checklists that documented adherence to the 200 min protocol structure, completion of all four scenario-based sessions, and accurate implementation of debriefing procedures, intervention fidelity was ensured. Two participants were scheduled to arrive at the designated location simultaneously, with each engaging in the IVRS session separately while maintaining an appropriate distance. After finishing the IVRS session, both participants underwent debriefing, with the trained research assistant serving as the facilitator for the IVRS and debriefing sessions. IVRS sessions were conducted with a 1:1 facilitator-to-participant ratio, whereas debriefing sessions were conducted with a 1:2 facilitator-to-participant ratio, following the GAS model [16]. After completing all four scenario-based IVRS practice and debriefing sessions, which lasted for 200 min, participants were advised to rest for 3 h given that VR-related symptoms may persist for 2–3 h [13]. This precautionary rest period was implemented to address potential cybersickness, a common side effect of prolonged VR exposure that can manifest as visual discomfort, nausea, dizziness, or eye strain [19]. The IVRS program involved moderate interaction within the virtual environment, including head movements for environmental observation and hand controller use for interface navigation, but did not require excessive physical movement or rapid motion that could exacerbate motion sickness symptoms. During the intervention, participants remained seated with back support and were monitored continuously for any signs of discomfort. The 3 h rest recommendation was based on established VR safety protocols and manufacturer guidelines, as cybersickness symptoms typically resolve within 2–3 h post-exposure for most individuals [19]. The research assistant conducted systematic symptom assessments using a standardized checklist (including fatigue, muscle pain, skin irritation, eye strain, and dizziness) both before and after each session, and only terminated the IVRS speak-up training session after confirming through direct questioning and observation that participants reported no adverse symptoms and appeared comfortable to leave independently.

### 2.7. Data Collection

Figure 4 presents a flow chart illustrating the data collection process. Data for the control and experimental groups were collected between June and October 2023 using self-administered questionnaires. Outcome variables were assessed at three time points. The pretest, conducted immediately before the intervention measured participants’ general characteristics, communication clarity, and collaborative attitudes. Post-test 1, administered immediately after the intervention, evaluated communication clarity and collaborative attitudes. Post-test 2, conducted 6 weeks post-intervention, assessed communication clarity, collaborative attitudes, and experiences of nursing malpractice.

### 2.8. Ethical Considerations

This study was approved by the Research Ethics Review Committee (IRB No. CR323014) of the Yonsei University Wonju Severance Christian Hospital. This ensured ethical compliance and protected the privacy of the study participants. Prior to participation, the research purpose, methods, intervention program, and principles of anonymity and confidentiality were explained to the participants. Moreover, the participants were informed that their involvement in the study was voluntary and that they could withdraw at any time without penalty. Written informed consent was obtained from all participants prior to data collection. The collected data were coded to maintain anonymity and stored on the researcher’s password-protected personal computer. Written documents were securely stored in a double-locked filing cabinet. Upon the completion of the study, a small token of appreciation (beverage coupon) was provided to all participants for their contribution.

### 2.9. Data Analysis

The collected data, which contained no missing values, were analyzed using the Statistical Package for the Social Sciences version 27.0 (IBM, New York, NY, USA). Descriptive statistics were used to analyze participants’ general characteristics. Homogeneity between groups was tested using the χ^2^ test, Fisher’s exact test, and independent t-test. Data normality was assessed using the Shapiro–Wilk test, revealing non-normal distributions for communication clarity (W = 0.94, *p* = 0.048) and collaborative attitudes (W = 0.91, *p* = 0.021). Group × time interaction effects were evaluated across three time points, using a generalized linear mixed model. Post hoc pairwise comparisons were conducted using Wilcoxon’s signed-rank test for within-group changes and Mann–Whitney *U* tests for between-group comparisons. Effect sizes were calculated with partial eta squared (η^2^*p*) for mixed models and *r* for nonparametric tests, using conventional interpretation criteria (small, medium, and large: 0.01, 0.06, and 0.14 for η^2^*p*; 0.10, 0.30, and 0.50 for *r*, respectively). Bootstrap methods were used for calculating 95% confidence intervals (95% Confidence Intervals, CIs). To account for multiple comparisons, we used the Benjamini–Hochberg procedure. A *p*-value of <0.05 indicated statistical significance.

## 3. Results

The reported results are based on unadjusted estimates, with 95% CIs provided to indicate the precision of the outcomes. Baseline comparison and homogeneity testing revealed no covariates as potential confounders. All continuous variables, including age and clinical experience, were analyzed as continuous measures without categorization.

### 3.1. Homogeneity Verification of the Experimental and Control Groups

All 36 enrolled novice nurses met the eligibility criteria and completed the study without dropout. Their data were then included in the final analysis, confirming the integrity of the sample. No significant differences were observed in the general characteristics (sex, age, clinical experience, and past VR learning experience) and baseline outcome variables (communication clarity and collaborative attitudes) between the experimental and control groups, confirming the homogeneity of the groups (Table 1).

### 3.2. Effectiveness Evaluation of the IVRS Speak-Up Training Program

The results of the effective evaluation of the IVRS speak-up training program are summarized in Table 2.

Communication clarity showed no significant interaction effect between groups and time points (F = 0.84, *p* = 0.437, η^2^*p* = 0.024), though the time effect was significant (F = 9.42, *p* < 0.001, η^2^*p* = 0.217), indicating that both groups improved over time. Significant improvements were observed in the experimental group from pretest to post-test 1 (Z = 2.70, *p* = 0.007) but were not sustained at 6 weeks (Z = 1.61, *p* = 0.108). Conversely, the control group showed sustained improvements at post-test 1 (Z = 2.59, *p* = 0.010) and post-test 2 (Z = 2.85, *p* = 0.004). Between-group comparisons revealed no statistically significant differences at any time point.

Collaborative attitudes demonstrated significant group differences (F = 9.22, *p* = 0.005, η^2^*p* = 0.213), time effects (F = 7.76, *p* = 0.001, η^2^*p* = 0.186), and importantly, a significant group × time interaction (F = 4.23, *p* = 0.020, η^2^*p* = 0.111). The experimental group showed significant improvements from pretest to both post-test 1 (Z = 3.31, *p* = 0.001) and post-test 2 (Z = 2.97, *p* = 0.003), demonstrating sustained positive effects. In contrast, the control group showed no significant improvement at post-test 1 (Z = 0.20, *p* = 0.842), with only marginal improvement at post-test 2 (Z = 1.88, *p* = 0.060). Between-group comparisons revealed significantly higher collaborative attitudes in the experimental group at post-test 1 (Z = 2.79, *p* = 0.005, *r* = 0.47) and post-test 2 (Z = 2.35, *p* = 0.019, *r* = 0.39).

No significant difference in nursing malpractice experience was observed between the experimental group (M = 0.50 ± 0.62) and control group (M = 0.72 ± 1.07) at 6 weeks post-intervention (Z = 0.16, *p* = 0.872, *r* = 0.03).

## 4. Discussion

This study examined the short- and long-term effectiveness of an IVRS speak-up training program for novice nurses. Adapted from a previously validated program for nursing students [15], this novel intervention incorporated structured communication strategies to enhance novice nurses’ ability to express concerns and convey clinical observations effectively. The findings demonstrated long-term improvements in collaborative attitudes (sustained at 6 weeks) and short-term improvements in communication clarity. Therefore, IVRS-based speak-up communication training may be useful in strengthening interprofessional teamwork and patient safety competencies during the early stages of nursing practice of novice nurses.

First, the experimental group showed a significant improvement in collaborative attitudes, which was maintained at 6 weeks post-intervention, demonstrating short- and long-term effects of the training. This finding underscores the effectiveness of IVRS speak-up training program in fostering interprofessional collaboration, which is essential for ensuring patient safety. The structured practice of mutually supportive communication strategies, particularly the two-challenge rule and the CUS technique, through four learning experience scenarios appears to have contributed meaningfully to the development of collaborative attitudes between novice nurses and physicians. This result aligns with previous findings demonstrating that immersive and interactive communication training can enhance interprofessional collaboration and team performance in healthcare settings [11,22,23,24].

Notably, the IVRS speak-up program provides advantages that are unique from those of traditional simulation approaches. It creates a psychologically safe learning environment, allowing novice nurses to practice assertive communication in 3D virtual spaces without fear of judgment from observers or instructors; this fear is a critical barrier to speaking up in hierarchical clinical settings [7,25]. Unlike conventional group-based training, participants in this program engaged in realistic hierarchical scenarios as IVRS that mirror actual nurse–physician power dynamics; thus, they could experience and practice speaking up in situations that closely replicate clinical reality. In addition, through exposure to multiple patient scenarios and complex interprofessional relationships, learners can navigate diverse communication challenges while receiving immediate feedback on the consequences of their communication choices [26]. Thus, this experiential learning approach may be particularly valuable for novice nurses with limited opportunities to engage in direct communication with physicians during the early stages of their clinical practice [7].

In particular, Liaw et al. [23] found that practicing structured and assertive communication methods such as ISBAR and the CUS technique in VR environments increased learners’ confidence and sense of team identity in interprofessional interactions. The substantial improvement in collaborative attitudes among the experimental group suggests that immersive and realistic situational contexts offered by IVRS fosters a better understanding with interprofessional collaboration [10]. As a supplemental modality to traditional group-based education and discussion, this training approach may help novice nurses enhance essential communication competencies, thereby promoting safer and more collaborative care environments.

Second, communication clarity showed no significant interaction effect between groups and time points, though improvements were observed in both groups immediately after the intervention. The pattern of short-term improvement observed among novice nurses aligns with the original IVRS program, which exhibited significant short-term improvements in team communication among undergraduate nursing students [15], and with previous studies demonstrating that assertiveness training through group lectures or discussions can effectively strengthen communication competencies [4,5,12,20]. Therefore, IVRS-based communication training has inherent characteristics that promote immediate skill acquisition but may require additional strategies to ensure long-term retention.

Remarkably, compared with the control group, the experimental group exhibited significantly greater improvements in communication clarity at post-test 1; however, this effect was not sustained at 6 weeks post-intervention. This decline indicates that the current single-session training approach, although it provided four multi-patient scenarios and was effective for immediate learning, may be insufficient for sustained behavioral change in complex clinical environments.

Alternatively, the observed decline may not reflect an actual reduction in communication ability but rather an increase in self-criticism. Given that this study utilized a self-assessment version of the Communication Clarity Scale [20] instead of the original observer-rated tool [13], the results may be particularly susceptible to changes in participants’ self-assessment standards. Increased confidence in interprofessional communication following IVRS education [23] may have led participants to adopt more rigorous criteria for assessing their own performance through this self-assessment format. In particular, the immersive nature of IVRS, supported by real-time feedback that highlights the immediate consequences of communication choices, may have enhanced participants’ metacognitive awareness of communication effectiveness, leading them to apply stricter self-evaluation standards for communication clarity. Furthermore, engaging in assertive communication across four realistic virtual scenarios may have developed participants’ critical self-reflection abilities, prompting more rigorous self-assessment during the follow-up period.

To address these limitations, nurses in future interventions should provide repetitive or booster training sessions at strategic intervals for novice ones before the 6-week mark to maintain initial gains. Spaced repetition and distributed practice have been reported to significantly enhance long-term retention of complex skills [8]. To sustain communication competencies over time, IVRS speak-up training may be combined with other evidence-based educational modalities, such as high-fidelity simulation training [16], mentorship programs [7], or reflective practice components [27]. Notably, IVRS speak-up training using an HMD in a 3D virtual space offers unique benefits that merit further exploration, particularly its ability to simulate complex multi-patient scenarios and intricate interprofessional dynamics that would be difficult to replicate in traditional simulation settings [26]. This modality ensures learner safety and supports repeated, immersive practice without time or physical space constraints.

In addition to its educational benefits, the IVRS program demonstrated notable cost-effective scalability compared to traditional high-fidelity simulation approaches [28]. The initial set up cost included Oculus Quest 2 HMDs, Unity 3D software licensing, and one-time development expenses for scenario programming. However, once established, the system requires minimal ongoing costs for maintenance and can accommodate multiple training sessions without additional material expenses or specialized facility requirements [29]. Unlike traditional simulation that necessitates expensive mannequins, dedicated simulation laboratories, and multiple trained facilitators [30], the IVRS approach requires only standard classroom space and one trained research assistant per session. This cost-effectiveness, combined with the program’s ability to provide consistent, standardized training experiences regardless of location or time constraints [28], makes it particularly attractive for healthcare institutions seeking to implement comprehensive communication training programs for novice nurses.

Lastly, the current study found no statistically significant differences in the frequency of nursing malpractice experiences between the experimental and control groups. This outcome may be attributed to the short 6-week evaluation period for the novice nurses included in this study, which may be insufficient to observe measurable changes in the frequency of malpractice experience. Additionally, novice nurses may require more time and practical opportunities to fully integrate assertive communication behaviors into their clinical practice [4,6]. Nevertheless, considering previous studies showing that effective team communication and positive collaborative attitudes can prevent medical errors [12,23], the IVRS speak-up program developed in this study, which provides a highly immersive and repeatable learning experience, is expected to be essential for patient safety by helping reduce potential nursing errors among novice nurses in the future.

The present study highlights important implications for interprofessional communication education, demonstrating the utility of IVRS as a transformative educational tool for novice nurses. By integrating standardized communication techniques (ISBAR and check-back) with mutually supportive communication techniques (two-challenge rule and CUS) through four multi-patient scenarios in an immersive educational environment, the program provides an effective learning experience that addressed limitations of traditional group education and discussions, thereby enhancing novice nurses’ speak-up communication skills for patient safety. A unique advantage of the program is its capacity to create psychologically safe environments, allowing learners to practice assertive communication without external observation, navigate realistic hierarchical and multi-patient scenarios, and receive immediate feedback on their communication choices. Our results emphasize that IVRS speak-up training can compensate for the limitations of high-fidelity simulation training, which requires the construction of a simulation training environment and a skilled instructor [15,23]. This advantage suggests the potential of the IVRS speak-up training program to complement existing educational modalities by providing realistic, risk-free environments that enhance novice nurses’ competencies while facilitating flexible implementation of booster sessions when needed for sustained skill acquisition over time. Furthermore, our results showed that learning assertive communication for patient safety through the IVRS speak-up training program effectively promoted sustained positive improvements in collaborative attitudes perceived by novice nurses. This finding demonstrates the value of using the IVRS speak-up training program as a self-paced, repeatable training program for improving the capabilities of novice nurses who may experience difficulties in demonstrating interprofessional teamwork necessary to maintain patient safety in clinical environments. Given the increasing emphasis placed by healthcare systems on interprofessional collaboration, the IVRS speak-up training program for patient safety is expected to prepare novice nurses for interprofessional collaboration.

The present study has several limitations that warrant acknowledgment. First, the relatively small sample size and quasi-experimental design involving non-equivalent groups may restrict generalizability and introduce potential threats to internal validity. Future research should employ randomized controlled trial designs with larger and more heterogeneous participant pools. Second, reliance on self-reported data may have introduced social desirability or recall bias. Future studies should incorporate observational assessments by trained raters to yield more objective insights into intervention effectiveness. Third, a significant limitation of this study lies in the comparison between IVRS and traditional lecture-based discussion format, which represents substantially different educational modalities. While this comparison demonstrated the superiority of active, immersive learning over passive educational approaches, it may be considered somewhat predictable given the extensive literature already establishing the advantages of simulation-based learning over traditional lecture formats. The marked difference between these two educational approaches may have contributed to the significant outcomes observed, potentially limiting the specificity of insights regarding IVRS effectiveness compared to other active learning strategies. Future research should prioritize direct comparisons between IVRS and traditional high-fidelity simulation to determine unique virtual reality benefits. Additional promising directions include longitudinal studies examining training durability beyond 6 weeks; comparative effectiveness research with emerging technologies; dose–response investigations; and cost-effectiveness analyses. These studies would provide comprehensive evidence for IVRS applications while addressing current methodological limitations.

## 5. Conclusions

In this quasi-experimental study conducted at a single tertiary hospital, the IVRS speak-up training program demonstrated effectiveness in enhancing assertive communication skills among the 36 novice nurses who participated. Within this sample, the results suggest that the program contributed to short-term improvements in communication clarity and fostered sustained improvements in collaborative attitudes, which are critical challenges for novice nurses navigating hierarchical clinical contexts. The sustained enhancement in collaborative attitudes lasting for 6 weeks post-intervention observed in this study population particularly highlights the program’s potential for impact on interprofessional relationships in similar clinical settings. In addition, given the study’s single-site design and convenience sampling approach, these findings should be interpreted within the context of the specific participant characteristics and institutional environment studied. For healthcare institutions with similar organizational structures and novice nurse populations, integrating this type of IVRS training into traditional communication education may serve as a potentially meaningful strategy to strengthen interprofessional collaboration and advance patient safety in clinical practice.

## Figures and Tables

**Figure 1 healthcare-13-02435-f001:**
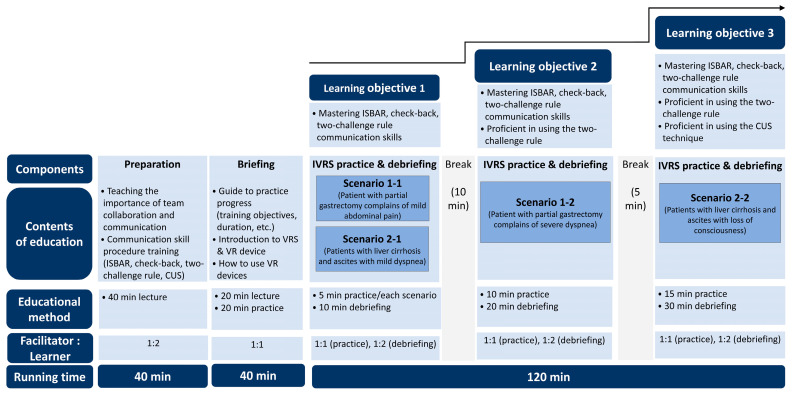
Structure of the IVRS speak-up training program. CUS = concerned, uncomfortable, safety issue; ISBAR = identification situation, background, assessment, recommendation; IVRS = immersive virtual reality simulation; VR = virtual reality; VRS = virtual reality simulation.

**Figure 2 healthcare-13-02435-f002:**
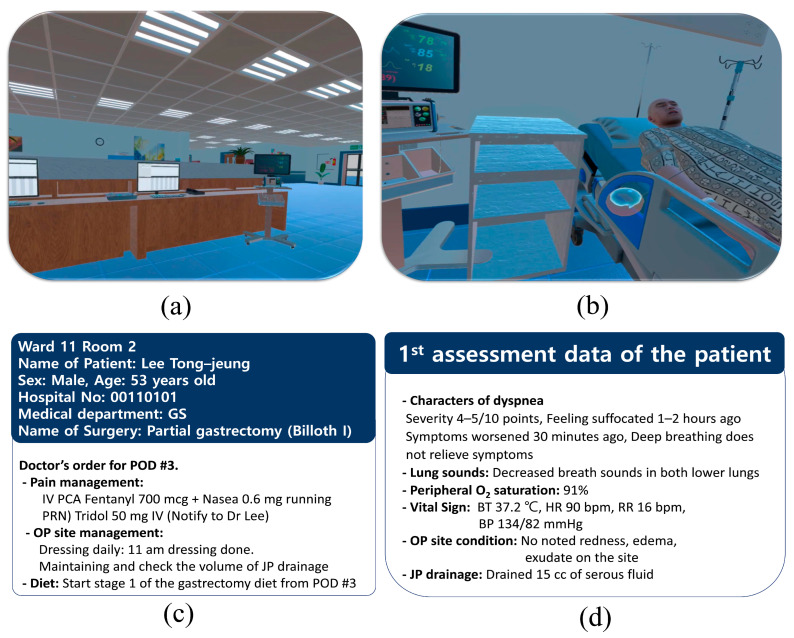
Screens of a virtual hospital, a patient, and the patient data provided. (**a**) Nurse’s station. (**b**) Patient’s room. (**c**) Patient’s information and physician’s orders. (**d**) Initial assessment data of the patient. BP = blood pressure; bpm = beat per minute; BT = body temperature; GS = general surgery; HR = heart rate; IV = intravenous; ISBAR = identification situation, background, assessment, recommendation; JP = Jackson-Pratt; No. = number; OP = operation; O_2_ = oxygen; PCA = patient-controlled analgesia; POD = postoperative day; PRN = as needed; RR = respiration rate.

**Figure 3 healthcare-13-02435-f003:**
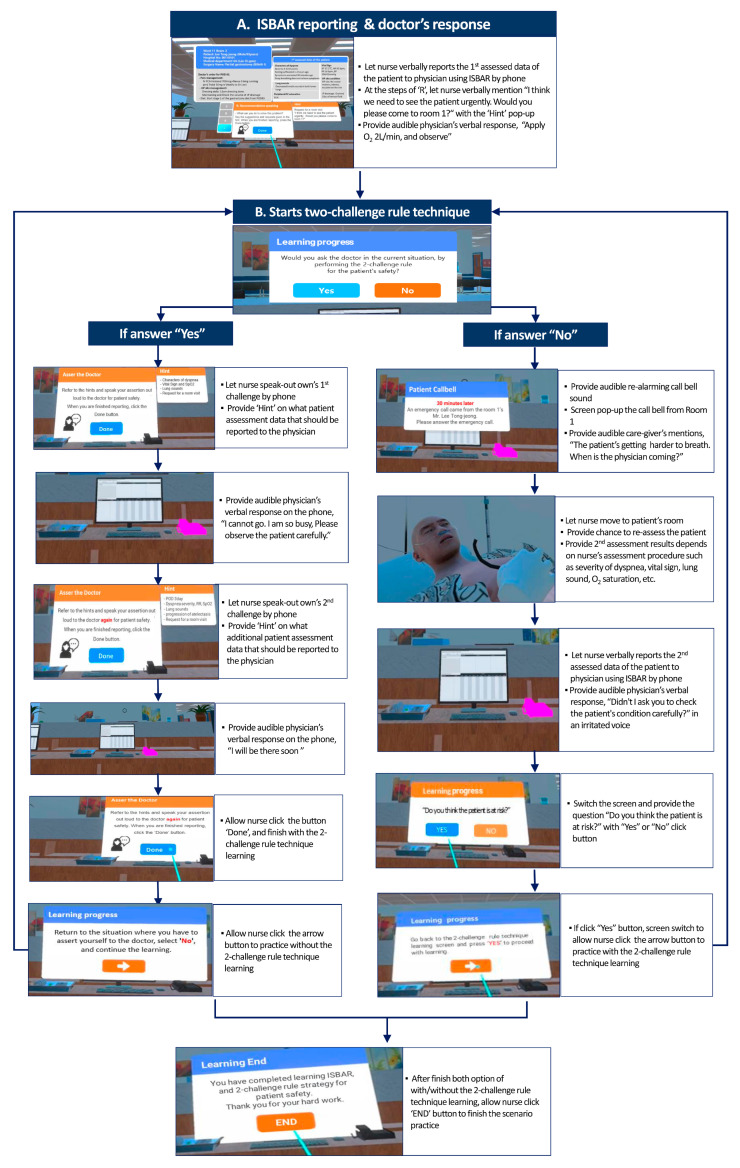
Operational workflow of the two-challenge rule in learning objective 2 of scenarios 1-2.

**Figure 4 healthcare-13-02435-f004:**
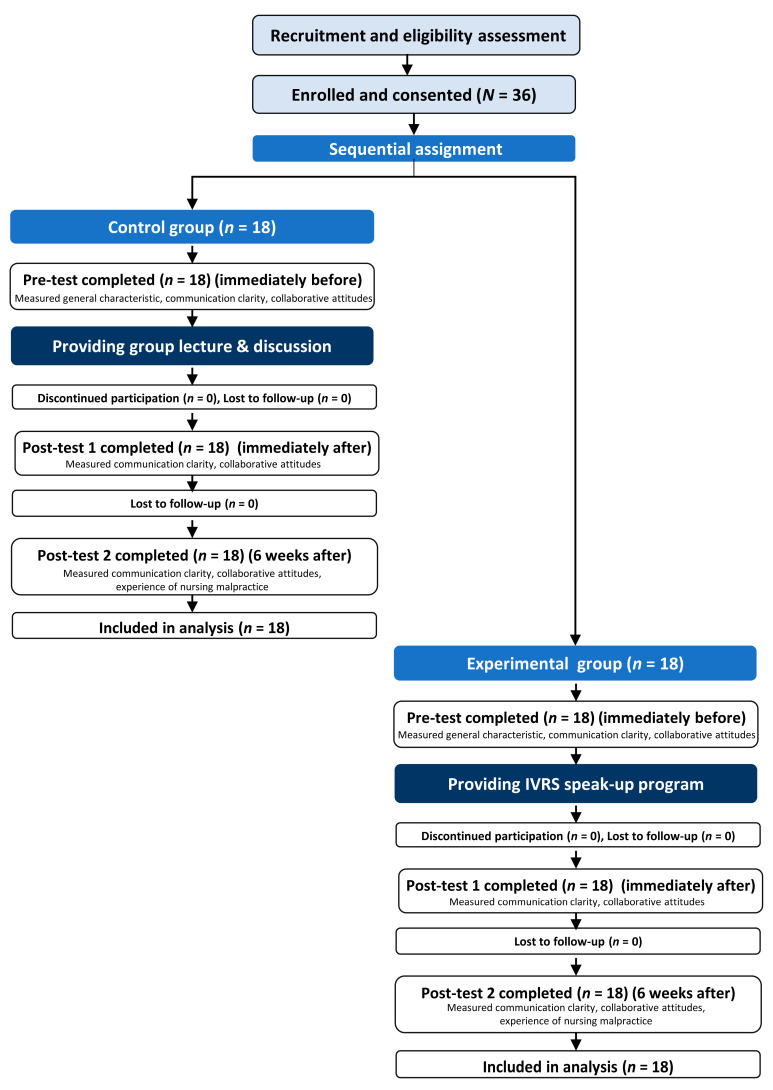
Flow chart of the data collection process in this quasi-experimental study. IVRS = immersive virtual reality simulation.

**Table 1 healthcare-13-02435-t001:** Homogeneity of the experimental and control groups (*N* = 36).

Variables	Categories	Experimental Group (*n* = 18)	Control Group (*n* = 18)	t/χ^2^ (*p*)
		*n* (%) or M ± SD		
General characteristics				
Sex	Male	3 (17)	6 (33)	1.33 (0.443) ^a^
	Female	15 (83)	12 (67)	
Age (years)		23.39 ± 1.46	24.72 ± 2.99	1.70 (0.101)
Clinical experience (months)		2.61 ± 0.71	1.94 ± 1.00	1.42 (0.166)
VR learning experience	Yes	4 (22)	5 (28)	0.15 (1.000) ^a^
	No	14 (88)	13 (72)	
Baseline outcome variables				
Communication clarity		4.21 ± 0.59	4.13 ± 0.49	0.46 (0.646)
Collaborative attitudes		3.58 ± 0.22	3.44 ± 0.29	1.57 (0.126)

M = mean; SD = standard deviation; VR = virtual reality. ^a^ Fisher’s exact test.

**Table 2 healthcare-13-02435-t002:** Effectiveness of the IVRS Speak-Up training program (*N* = 36).

Outcome Variables	Groups	Pretest	Post-Test 1	Post-Test 2	Post-Test 1: Pretest	Post-Test 2: Pretest	Source	F (*p*) η_2_*p* [95% CI]
		M ± SD			Z (*p*) [95% CI]	Z (*p*) [95% CI]		
Communication clarity	Experimental (*n* = 18)	4.21 ± 0.59	4.64 ± 0.48	4.42 ± 0.43	2.70 (0.007) ^a^ [0.12, 0.74]	1.61 (0.108) ^a^ [–0.04, 0.46]	G	0.31 (0.580) 0.009 [0.000, 0.112]
	Control (*n* = 18)	4.13 ± 0.49	4.46 ± 0.49	4.47 ± 0.53	2.59 (0.010) ^a^ [0.08, 0.58]	2.85 (0.004) ^a^ [0.11, 0.57]	T	9.42 (<0.001) 0.217 [0.078, 0.380]
	Between-group comparison Z (*p*), r	0.43 (0.668) ^b^, 0.07	1.27 (0.203) ^b^, 0.21	0.68 (0.495) ^b^, 0.11			G × T	0.84 (0.437) 0.024 [0.000, 0.141]
Collaborative attitudes	Experimental (*n* = 18)	3.58 ± 0.22	3.78 ± 0.17	3.75 ± 0.17	3.31 (0.001) ^a^ [0.08, 0.32]	2.97 (0.003) ^a^ [0.06, 0.29]	G	9.22 (0.005) 0.213 [0.047, 0.380]
	Control (*n* = 18)	3.44 ± 0.29	3.44 ± 0.37	3.56 ± 0.26	0.20 (0.842) ^a^ [–0.12, 0.12]	1.88 (0.060) ^a^ [–0.01, 0.25]	T	7.76 (0.001) 0.186 [0.055, 0.343]
	Between-group comparison Z (*p*), r	1.48 (0.139) ^b^, 0.25	2.79 (0.005) ^b^, 0.47	2.35 (0.019) ^b^, 0.39			G × T	4.23 (0.020) 0.111 [0.007, 0.264]
Experience of nursing malpractice	Experimental (*n* = 18)			0.50 ± 0.62				
	Control (*n* = 18)			0.72 ± 1.07				
	Between-group comparison Z (*p*), r			0.16 (0.872) ^b^, 0.03				

G = Group; G × T = group × time interaction; T = time; M = mean; SD = standard deviation; CI = confidence interval; IVRS = immersive virtual reality simulation. η^2^*p* = partial eta squared (small: 0.01, medium: 0.06, large: 0.14). r = effect size for Mann–Whitney U test (small: 0.10, medium: 0.30, large: 0.50). Statistical tests: ^a^ Wilcoxon’s signed-ranks test; ^b^ Mann–Whitney U test. Pretest = immediately before the intervention; Post-test 1 = immediately after the intervention; Post-test 2 = 6 weeks after the intervention.

## Data Availability

The data presented in this study are available from the corresponding author upon reasonable request. The data are not publicly available due to privacy restrictions.

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
