# Peer review of "Effectiveness of an Immersive Virtual Reality Simulation Speak-Up Training Program for Patient Safety in Novice Nurses: A Quasi-Experimental Study"

_healthcare, 2025, doi:10.3390/healthcare13192435_

Round 1
Reviewer 1 Report
Comments and Suggestions for Authors
General Comment
This is an interesting and relevant study that evaluates the use of immersive virtual reality training for improving communication and collaboration in patient safety. The topic is timely and of practical significance for healthcare education. However, there are several clarifications and improvements required to enhance clarity, transparency, and scientific rigor.
Abstract
- The abstract would be more understandable if it included details about the control group’s intervention.
- Introduction
“Integrating VRS with structured communication frameworks, such as TeamSTEPPS program, which incorporates evidence-based communication techniques (e.g., ISBAR, check-back, two-challenge rule, and CUS)”
The acronyms should be written out in full, and to improve understanding for all healthcare professionals, there should be a brief summary of what these communication techniques are and what they are used for: ISBAR, check-back, two-challenge rule, and CUS.
- Materials and Methods
2.2.2. Description of the IVRS speak-up training program
“The IVRS speak-up training program was structured into three phases (preparation, briefing, IVRS practice and debriefing of four multipatient scenarios)”
There is an inconsistency: the text states three phases but lists four. This should be clarified.
Figure 1: Requires better spacing for improved readability.
It would also be valuable to provide details on the methodology used with the control group for proper comparison.
2.3. Participants and setting
“as those under certain medications”
This is vague and needs clarification. What medications were considered exclusion criteria?
2.4.1. Communication clarity
Who assessed the Communication clarity scale? Please specify the evaluators.
2.4.3 Experience of nursing malpractice
Who assessed the Experience of nursing malpractice?
Additionally, clarify how this variable relates to the intervention.
2.7. Data collection
“Predefined dropout criteria included withdrawal of consent, occurrence of severe VR-related symptoms (e.g., persistent nausea, dizziness, or eye strain lasting >30 min), inability to complete the full intervention protocol, and failure to attend scheduled sessions. Within the study period, no participants met either of these criteria.”
This information would be more appropriate under Section 2.3 (Participants and setting).
- Discussion
“Alternatively, the observed decline may not reflect an actual reduction in communication ability but rather an increase in self-criticism. Increased confidence in interprofessional communication following IVRS education [25] may have led participants to adopt more rigorous stringent criteria for assessing their own performance.”
There is a issue here. The scale used to assess communication, based on the reference provided (Marshall, S.; Harrison, J.; Flanagan, B. The teaching of a structured tool improves the clarity and content of interprofessional clinical communication. Qual. Saf. Health Care 2009, 18, 137–140, doi:10.1136/qshc.2007.025247), is not a self-assessment scale. This needs clarification, as the interpretation of results may be misleading.
- Conclusions
Conclusions should reflect the limitations of the study and therefore be restricted to the sample studied, avoiding generalizations.
The statement:
“Informed Consent Statement: Written informed consent has been obtained from the patient(s) to publish this paper.”
This should be corrected, as consent was obtained from participants, not patients.
Author Response
Thank you very much for taking the time to review this manuscript. Please find the detailed responses below and the corresponding revisions in red in the re-submitted files.

Reviewer 2 Report
Comments and Suggestions for Authors
This study was sound in the design and implementation, but I had one concern regarding the comparison in the study. Comparing Immersive Virtual Reality Simulation (IVRS) to a lecture based education program seems academic. Many studies have been done comparing lecture to forms of simulation, and it's been shown that active learning strategies like traditional simulation, virtual reality, and so forth. So, it makes sense that there was a significant change between lecture discussions and IVRS since the two modalities are very different. But certainly comparing traditional simulation to IVRS could offer some other insights that could be more profound. I think in the limitations this needs to be mentioned that the comparison is substantially different. Also, there needs to be mention of future implications in the discussion, such as comparing traditional simulation, such as high fidelity simulators, with IVRS. Providing a section specifically on other ways you would like to take this study would be helpful.
I did have concerns about the amount of time used to allow participants to rest after IVRS due to the effects of the IVRS. It wasn't clear how much interaction would be involved. Was there excessive movement? I think the duration of rest an concerns about cybersickness need to be explained more.
I do feel these are areas of the design that need further discussion and explanation to help this study be better received. Another thing that might could assist is a brief understanding of the cost involved in setting up the IVRS, which I don't believe is mentioned.
Author Response

(The authors gave the same response as above.)

Round 2
Reviewer 2 Report
Comments and Suggestions for Authors
Thank you for addressing the concerns I raised about the study design. Adding that to the limitations section shows you understand this has been studied extensively and could yield predictable results. I believe the changes you have made show the rigor you've been willing to put back into the manuscript to make it publication-worthy.